# Natural arbovirus infection rate and detectability of indoor female *Aedes aegypti* from Mérida, Yucatán, Mexico

**Oscar David Kirstein**[1ʘ], **Guadalupe Ayora-Talavera**[2ʘ], **Edgar Koyoc-Cardeña**[3], **Daniel Chan Espinoza**[3], **Azael Che-Mendoza**[3], **Azael Cohuo-Rodriguez**[3], **Pilar Granja-Pérez**[4], **Henry Puerta-Guardo**[3], **Norma Pavia-Ruz**[5], **Mike W. Dunbar**[1], **Pablo Manrique-Saide**[3], **Gonzalo M. Vazquez-Prokopec**[1] *

**1** Department of Environmental Sciences, Emory University, Atlanta, Georgia, United States of America, **2** Laboratorio de Virología. Centro de Investigaciones Regionales "Dr. Hideyo Noguchi", Universidad Autónoma de Yucatán, Mérida, Yucatán, México, **3** Unidad Colaborativa de Bioensayos Entomológicos, Campus de Ciencias Biológicas y Agropecuarias, Universidad Autónoma de Yucatán, Mérida, Yucatán, México, **4** Laboratorio Estatal de Salud Pública, Servicios de Salud de Yucatán, Mérida, Yucatán, México, **5** Laboratorio de Hematología. Centro de Investigaciones Regionales "Dr. Hideyo Noguchi", Universidad Autónoma de Yucatán, Mérida, Yucatán, México

ʘ These authors contributed equally to this work.
* gmvazqu@emory.edu

## Abstract

Arbovirus infection in *Aedes aegypti* has historically been quantified from a sample of the adult population by pooling collected mosquitoes to increase detectability. However, there is a significant knowledge gap about the magnitude of natural arbovirus infection within areas of active transmission, as well as the sensitivity of detection of such an approach. We used indoor *Ae. aegypti* sequential sampling with Prokopack aspirators to collect all mosquitoes inside 200 houses with suspected active ABV transmission from the city of Mérida, Mexico, and tested all collected specimens by RT-PCR to quantify: a) the absolute arbovirus infection rate in individually tested *Ae. aegypti* females; b) the sensitivity of using Prokopack aspirators in detecting ABV-infected mosquitoes; and c) the sensitivity of entomological inoculation rate (EIR) and vectorial capacity (VC), two measures ABV transmission potential, to different estimates of indoor *Ae. aegypti* abundance. The total number of *Ae. aegypti* (total catch, the sum of all *Ae. aegypti* across all collection intervals) as well as the number on the first 10-min of collection (sample, equivalent to a routine adult aspiration session) were calculated. We individually tested by RT-PCR 2,161 *Aedes aegypti* females and found that 7.7% of them were positive to any ABV. Most infections were CHIKV (77.7%), followed by DENV (11.4%) and ZIKV (9.0%). The distribution of infected *Aedes aegypti* was overdispersed; 33% houses contributed 81% of the infected mosquitoes. A significant association between ABV infection and *Ae. aegypti* total catch indoors was found (binomial GLMM, Odds Ratio > 1). A 10-min indoor Prokopack collection led to a low sensitivity of detecting ABV infection (16.3% for detecting infected mosquitoes and 23.4% for detecting infected houses). When averaged across all infested houses, mean EIR ranged between 0.04 and 0.06 infective bites per person per day, and mean VC was 0.6 infectious vectors generated from a population feeding on a single infected host per house/day. Both measures were

**Data Availability Statement:** All relevant data are within the manuscript and its Supporting Information file.

**Funding:** Research funding was provided by an Interagency Agreement between USAID and the US Centers for Disease Control and Prevention (CDC: OADS BAA 2016-N-17844; GMVP, PI), by the Canadian Institutes of Health Research (CIHR) and IDRC (Preventing Zika disease with novel vector control approaches, Project 108412; PMS, PI) by Fondo Mixto CONACyT (Mexico)-Gobierno del Estado de Yucatán (Project YUC-2017-03-01-556; PMS, PI), the National Institutes of Health (NIH/ NIAID: U01AI148069; GMVP, PI) and Emory University via the MP3 initiative (GMVP, PI). The opinions expressed herein are those of the author (s) and do not necessarily reflect the views of Emory University or the U.S. Agency for International Development. The findings and conclusions in this paper are those of the authors and do not necessarily represent the official position of the Centers for Disease Control and Prevention. The funders had no role in study design, data collection and analysis, decision to publish, or preparation of the manuscript.

**Competing interests:** The authors have declared that no competing interests exist.

significantly and positively associated with *Ae. aegypti* total catch indoors. Our findings provide evidence that the accurate estimation and quantification of arbovirus infection rate and transmission risk is a function of the sampling effort, the local abundance of *Aedes aegypti* and the intensity of arbovirus circulation.

## Author summary

*Aedes*-borne diseases comprise a serious public health burden in many parts of the world, usually affecting low income areas. The ability to detect virus circulation within a population may be key in responding to the threat of outbreaks, providing a cost-effective approach for triggering vector control. Unfortunately, gaps in the knowledge of natural *Aedes*-borne virus (ABV) infection in *Aedes aegypti* have led to uncertainties in the consideration of arbovirus surveillance in mosquitoes. Here, we show that the natural infection rate in a mosquito population may not be a function of where *Aedes aegypti* are, but rather where key human-mosquito contacts occur. Sampling 200 houses with suspected ABV active transmission led us to quantify high virus infection rates in all *Aedes aegypti* present in the house and use such information to estimate the sensitivity of indoor aspiration with Prokopack devices and two measures of ABV transmission potential. Our findings provide evidence that the accurate quantification of arbovirus infection rate and transmission risk is a function of the sampling effort, the local abundance of *Aedes aegypti* and the intensity of arbovirus circulation. Results from this study are relevant to understand the value of virus testing of vector populations, and for the design of entomological endpoints relevant for epidemiological trials quantifying the impact of vector control on ABVs.

## Introduction

Emerging *Aedes*-borne viruses (ABVs) such as chikungunya (CHIKV), Dengue (DENV) and Zika (ZIKV) contribute significantly to the global burden of infectious diseases [1–3]. Transmitted primarily by the ubiquitous and highly anthropophilic mosquito *Aedes aegypti*, these viruses have propagated throughout tropical and subtropical urban environments often co-circulating within the same period and geographical areas [4–8]. Infections of CHIKV, DENV and ZIKV can present similar symptoms, ranging from asymptomatic to mild or inapparent to severe illness with life-threatening manifestations and death [6, 9]. ZIKV and CHIKV infections, particularly in the Americas, have been linked to fetus abnormalities during pregnancy, neurological complications, o chronic joint diseases in adults that can persist for years [10, 11]. The co-circulation of arboviral infections and their epidemic propagation challenge differential diagnoses, primary patient care, and limit the effectiveness of existing vector control tools [5, 8, 12–15]. Furthermore, the lack of accurate entomological correlates of ABV risk [2, 16, 17], is affected by multiple sources of bias including the difficulty of detecting and accurately quantifying immature or adult *Ae. aegypti* density [18], the exposure of people to mosquitoes in residences other than their homes [19, 20], the variable level of susceptibility in the human population against each virus [21], or the limited predictive power of entomological indices for informing vector control [22].

*Aedes aegypti* is considered a very efficient vector of ABVs even at low apparent population densities [23, 24]. A common assumption in ABV research is that due to the low vector density

and focal nature of human-mosquito contacts [19], natural arbovirus infection in *Ae. aegypti* is very low [25, 26], limiting the implementation of entomo-virological surveillance systems as conducted for other urban arbovirus (e.g., West Nile virus [27]). The quantification of infection rates in mosquito populations depends on the methodology used to detect viral infection. Methods for virus detection include cell culture [28, 29], immunoassay [28, 30] or molecular methods [5, 8, 31]. Reverse transcription–polymerase chain reaction (RT-PCR) followed by amplicon sequence is considered the benchmark for infection confirmation and virus discrimination. Given processing costs, and often limited mosquito yields, ABV detection tends to be conducted in pools of mosquitoes, generally between 10 and 20 individuals per pool [27]. In the presence of focal transmission (e.g., multiple infected mosquitoes within a single premise, infecting many individuals), such pooling method may lead to bias in the estimation of ABV natural infection rates [32, 33]. Part of this bias is introduced by the calculation of the minimum infection rates (MIR) and the maximum likelihood rate (MLR), which make different assumptions about the frequency and aggregation of infection rates, but that are not sensitive to extreme variability in the distribution of infected mosquitoes [27, 33, 34].

Despite these assumption and limitations, multiple research groups have quantified infection rates in *Ae. aegypti* with different levels of success. ABV entomo-virological characterization in *Ae. aegypti* from northern Brazil detected only 7 out of 37 pools (containing 10 mosquitoes each) tested and ~1000 mosquitoes collected [8]. A study conducted during the DENV transmission peak in Mérida, Mexico, found that after individually testing *Ae. aegypti* mosquitoes only 66 females out of 10,254 (<1%) were positive for DENV [30]. These findings outline a common issue with population-wide cross-sectional quantifications of ABV infection: the natural infection rate of an *Ae. aegypti* population may not be a function of where *Ae. aegypti* are, but rather where key human-mosquito contacts occur [35]. The possibility for early detection of virus circulation within a population may be key in preventing outbreaks, providing a cost-effective approach for triggering vector control. In a study conducted in Guerrero, Mexico, circulation of CHIKV within mosquito populations was detected 10 days prior to any reported symptomatic human case, which allowed for early vector control actions and outbreak mitigation [7, 36].

The capacity of capturing a considerable and representative sample of mosquitoes is necessary for a comprehensive characterization of their natural infection. A myriad of adult *Ae. aegypti* sampling methods have been used for quantifying ABV natural infection rate. While passive traps (BG sentinel, sticky ovitraps, Gravid *Aedes* Traps (GAT), autocidal *Aedes* gravid ovitrap [37]) may allow for widespread coverage, they also require multiple days for capturing enough mosquitoes for virus testing and their sensitivity to vector and virus detection is unknown. On the positive side, passive traps do not require premise entry (an issue currently in the COVID-19 pandemic) and have been used to detect ABV-infected *Ae. aegypti* (e.g., [38]). Adult aspiration, while it is assumed to be more laborious and dependent on trained staff, provides an instantaneous measure of vector density and is considered a gold standard for adult *Ae. aegypti* collection [37, 39]. Applying sequential removal sampling using Prokopack aspirators [18, 39] the absolute density of *Ae. aegypti* was found to be up to five times bigger than previously estimated implementing the standard 10-minute collection period per household. As all studies quantifying ABV infection in *Ae. aegypti* have sampled a small fraction of the adult population and pooled collected mosquitoes to increase yield and detectability, there is a significant knowledge gap with regards to the magnitude of natural ABV infection rates within areas of active transmission.

There is a need for improving the evidence base of the epidemiological impact of vector control on ABV [40]. Estimates of ABV infection in *Ae. aegypti* infection could be calculated as measures of intervention impact, provided they are accurately quantified (e.g., [23]). In

preparation for a clinical trial evaluating the epidemiological impact of targeted indoor residual spraying (TIRS) on ABVs [41], here we extended an observational study that used exhaustive Prokopack collections in houses with suspected active virus transmission [18] to quantify absolute ABV infection rate in individual *Ae. aegypti*. Specifically, we quantified: a) how arbovirus infection in *Ae. aegypti* (overall and for each virus separately) is distributed across houses with suspected ABV transmission; b) the association between ABV infection in *Ae. aegypti* and different measures of indoor adult vector density; c) the sensitivity of indoor adult *Ae. aegypti* collections using Prokopack aspirators in detecting ABV-positive mosquitoes; and d) effect of imperfect sampling of the adult population and vector density on two entomological measures of virus transmission potential, the entomological inoculation rate [42] and vectorial capacity [43].

## Material and methods

### Ethics statement

Protocols for this study were approved by Emory University's ethics committee under protocol ID: IRB00082848. The protocol was also approved by the Ethics and Research Committee from the O´Horan General Hospital from the state Ministry of Health, Register No. CEI-0-34-1-14. Written informed consent was obtained from the head of household prior to mosquito collection.

### Study area and design

The study was conducted in Mérida (population ~1 million), Yucatán, Mexico. Mérida is endemic for dengue [3, 4, 44] and, as most of the Americas, was recently and sequentially invaded by CHIKV and ZIKV [14]. Arbovirus transmission is seasonal, peaking during the rainy season (July-November). Since 2011, Mérida is home of a longitudinal cohort study called "Familias sin Dengue" (FSD, Families without dengue) that has characterized arbovirus infection and seroconversion rates and the entomological correlates of dengue infection [3, 4, 44]. Our study design originally involved selecting a total of 200 houses within FSD city blocks where recent (within one month) CHIKV, ZIKV or DENV occurred [18]. Given the low number of symptomatic ABV cases detected by FSD in 2015 (8 ZIKV, 12 DENV, 30 CHIKV [4], we modified our protocol by focusing on passive surveillance data collected by the Yucatán Ministry of Health (MOH) to achieve our target of 200 houses during the 2016–2017 seasons. Such dataset was not included in the FSD IRB and the level of masking in the data was determined by Yucatán MOH. While Yucatán MOH provided information for each virus geocoded to the census tract level [14], we could not obtain information that could identify which virus each household was positive to. Given the protocols for human subjects and household access, the team received a list of houses without information of how many individuals were infected (or when onset of symptoms occurred) or the virus infecting them. Therefore, the entomological team only had a list of houses to visit, and they were blind to any information about arbovirus infection status or intensity in each house. Collections occurred on 2016 and 2017 and concentrated during the period of ABV transmission (June to December). DENV and CHIKV were reported to the city's passive surveillance system in 2015, and ZIKV was first reported in 2016 (S1 Fig).

After obtaining informed consent from householders, exhaustive adult mosquito collections with Prokopack aspirators [39] were conducted using removal sampling, as described by Koyoc-Cardeña et al. [18]. Briefly, trained fieldworkers sequentially entered each house and collected mosquitoes from each room (including the kitchen and bathroom). Removal sampling was conducted with a constant effort at predefined intervals of 10 min over the course of

three hours or, if during two consecutive rounds no *Ae. aegypti* were captured. All personnel used regular field clothes, which included closed toe shoes, socks, pants and long-sleeve shirt, leaving little exposed skin. No DEET was used, and personnel captured any flying insect while moving throughout each room.

Collected mosquitoes were transported alive to the Autonomous University of Yucatán entomology lab (UCBE-UADY) and immobilized at −20°C for 10 min for sexing and taxonomical identification using standard keys. Additionally, blood-fed female *Ae. aegypti* were classified by the degree of blood digestion according to the Sella scale [45, 46], which was extended to include recent feeding as a category (the presence of bright red blood, regardless of its volume, was indicative of blood feeding within 24h of collection, and assigned a category '2' of Sella). Finally, male and female *Ae. aegypti* were individually dissected, their heads and bodies were separated and preserved in 1.5ml vials containing RNALater (Thermo Fisher Scientific, Waltham, MA, USA) with 1.5µl Tween 20 (Sigma-Aldrich Co.) and stored at -20°C for future virus detection by molecular methods.

### Detection of arboviral infections in *Ae. aegypti*

Initially, RNA was extracted from bodies (thorax, abdomen, and extremities). Individual specimens were homogenized using a cordless motor tissue distributor (Kimble) in a 1.5ml microcentrifuge tube with 150µl of PBS 1X, p.H 7.2 (GIBCO) and centrifuged at 4°C for 10 minutes at 1,500g. Total RNA was extracted from 140µl of the mosquito's body disruption supernatant using QIAamp Viral RNA Mini Kit (QIAGEN) following the manufacturer's recommendations. Finally, extracted RNA was eluted with 40µl of RNA-ase free water and preserved at -80°C. RNA extraction from heads was performed only from bodies that were positive for any of the targeted virus.

Detection of viral RNA was carried out by real-time RT-PCR using a probe-based detection method with a QuantiFast Probe RT-PCR Kit (QIAGEN). RT-PCR reactions were performed in a Step One Plus Real-Time PCR System (Applied Biosystems) following standard protocols. Reactions (samples) were considered positive when a sigmoidal curve was detected at a Ct value ≤38 cycles of amplification. S1 Table shows the Primers and probes used to target CHIKV, ZIKV [47, 48] and DENV (personal communication from Davis Arbovirus Research & Training).

Positive samples for CHIKV and ZIKV were reconfirmed by end-point RT-PCR using a high-fidelity polymerase, SuperScript III One-Step RT-PCR System with Platinum *Taq* DNA polymerase (Thermo Fisher Scientific). Primers were specifically designed to target a 420bp fragment of the viral gene E1 of CHIKV (including the M13 universal sequence, underlined): Fwd (5'–<u>TGTAAAACGACGGCCAGT</u>AGACGTCTATGCTAATACACAACTG—3') and Rev (5'–<u>CAAGAAACAGCTATGACC</u>TGAGAATTCCCTTCAACTTCTATCT—3'); or a fragment of 662 bp of the viral gene NS1 of ZIKV (primers were kindly provided by MSc. Jesus Reyes and are available upon request). PCR positive amplicons were sequenced for molecular confirmation of virus presence. For DENV, sequencing was performed on the amplicons obtained from the qRT-PCR, corresponding to a fragment of 212 bp of the NS5 viral gene. Samples with evidence of ABV infection by qRT-PCR were sent to Macrogen corp and sequenced by Sanger Method.

### Sequence analysis

Single forward and reverse raw sequencing data were assessed based on quality score. Reads were compared to those from the GenBank database using NCBI BLASTN (https://blast.ncbi.nlm.nih.gov/Blast.cgi) at default parameters (Madden 2013). BLAST "hits" were used to assign

reads to virus type, statistical significance was measured by the E-value and percentage or coverage. Reads that did not fulfill these conditions were considered potential chimeric sequences and discarded. Visualization of electropherograms, nucleotide sequences manipulation, alignment and analysis were performed using the software *Genious* Prime 2020.0.4 [49].

## Data analysis

In the context of this study, absolute *Ae. aegypti* density per house (termed 'total catch') was calculated as the sum of adult females collected across all sampling rounds, whereas relative density was calculated as the number of females per unit time (e.g., 10 minutes). We call this second measure the 'sample'. Total catch is an accurate estimate of *Ae. aegypti* absolute density indoors, as our prior study showed that both measures did not differ statistically from each other [18]. For analyses, houses were categorized based on their *Ae. aegypti* female total catch as high ($\geq$ 10 collected) or low (<10 collected), as in Koyoc-Cardeña et al. [18]. Absolute natural infection rate was calculated as the total number of infected females divided by the total catch per house, whereas relative natural infection rate was calculated as the number of infected and collected *Ae. aegypti* within a given unit of collection time (e.g., 10-minutes). The sensitivity of the adult aspiration to the detection of infected *Ae. aegypti* mosquitos was estimated by plotting the cumulative relative natural infection rate as a function of the collection time (catch effort). Chi-squared tests were used to compare infection rates by house, based on their density category (low vs high). To quantify the relationship between female adult *Ae. aegypti* density (count variable) and ABV infection (binary variable: infected = 1, not infected = 0) at the house level, generalized linear mixed models with a binomial link function and a random intercept associated with each house ID were employed, as described in Vazquez-Prokopec et al. [20]. The same model was extended to include other predictor variables, such as the presence of a blood meal in the mosquito (binary) or the Sella engorgement score of females (categorical) [50].

Two measures of ABV transmission potential were calculated using individual-level estimates of biting probability, infection, and vector density. The Entomological Inoculation Rate (EIR, expressed as the number of potentially infectious bites per person per day), routinely calculated for malaria [42], is considered a measure of human exposure to infectious mosquitoes. Vectorial capacity (VC) is a common metric that estimates the number of infectious vectors generated from a population feeding on a single infected host per unit area/time [43]. We calculated the EIR of ABVs at the household-level using the following equation: $IP = mas$; where *m* is the ratio of *Ae. aegypti* females to the number of residents of each house, *a* is the number of bites per day (calculated as the ratio of *Ae. aegypti* females with Sella's score 2 by the total number of *Ae. aegypti* females per house; Sella's score 2 indicates evidence of a bloodmeal within 24hs of capture) and *s* is the proportion of *Ae. aegypti* females found infected with any ABV.

We estimated the daily VC of ABVs per house, as follows: $VC = \frac{ma^2p^n}{-Ln(p)}$, where *m* and *a* are equivalent as in EIR and *p* is the daily survival probability of female mosquitoes (set as p = 0.7) and *n* daily probability of infection (set as n = 1/EIP, where EIP is the extrinsic incubation period; EIP = 5 days).

We calculated both EIR and VC for the total catch as well as the first round (sample) and conducted paired t-test to evaluate the difference in their value between both entomological measures by house. A GLMM with a Gaussian link function and random effect at the house level was applied to evaluate the association between each metric (EIR, VC set as dependent variables) and the total catch of *Ae. aegypti* by house.

All analyses were performed within the R programing environment (https://www.r-proje ct.org/) and GAMMs were run using the *lme4* package [34]. All original data used in this manuscript is included as S1 Data.

## Results

### Characteristics of ABV-infected *Ae. aegypti*

A total of 3,439 *Ae. aegypti* were collected in 179 houses, with 2,161 being females (62.8%). Of all collected females, 166 (7.7%) were positive for arbovirus infection (Table 1). The majority of infections were identified as positive for CHIKV (77.7%), followed by DENV (11.4%) and ZIKV (9.0%); coinfection with CHIKV and ZIKV was detected in three mosquitoes (1.8%) (Table 1). While the average number of female *Ae. aegypti* per house was very similar between 2016 and 2017 (12.9 and 12.7, respectively), the infection rate did differ significantly between years (16.4% in 2016 and 2% in 2017; $X^2$ = 152; $P<0.001$; Table 1). Interestingly, CHIKV was significantly more prevalent in 2016 than in 2017 (15.2% to 0%, respectively) whereas DENV was significantly less prevalent in 2016 than in 2017 (0.2% to 1.3%, respectively, $X^2$ = 6.7, $P$ <0.005, Table 1). ZIKV infection did not differ significantly between years (0.7% for both

**Table 1. Descriptive measures and Infection rates in indoor resting *Ae. aegypti* mosquitoes from Yucatán, Mexico, collected during the ABV transmission seasons of 2016 and 2017.**

| Entomologic measure | Collection Year | | Total |
|---|---|---|---|
| | **2016** | **2017** | |
| # of houses screened | 83 | 117 | 200 |
| # of infested houses with *Ae. aegypti* (% of infested houses) | 72 (86.7%) | 107 (91.4%) | 179 (89.5%) |
| # of infested houses with *Ae. aegypti* females (% of positive houses) | 66 (91.7%) | 103 (96.3%) | 169 (94.4%) |
| Total # of *Ae. aegypti* | 1,341 | 2,098 | 3,439 |
| # of *Ae. aegypti* females (% of females) | 851 (63.5%) | 1,310 (62.4%) | 2,161 (62.8%) |
| # of *Ae. aegypti* males (% of males) | 490 (36.5%) | 788 (37.5%) | 1,278 (37.2%) |
| Sex ratio F:M | **1.7:1** | **1.7:1** | **1.7:1** |
| # of positive *Ae. aegypti* females for any virus (% female tested) | 140 (16.4%) | 26 (2.0%) | 166 (7.7%) |
| # of positive *Ae. aegypti* females for CHIKV (% female tested) | 129 (15.2%) | 0 (0.0%) | 129 (6.0%) |
| # of positive *Ae. aegypti* females for DENV (% female tested) | 2 (0.2%) | 17 (1.3%) | 19 (0.9%) |
| # of positive *Ae. aegypti* females for ZIKV (% female tested) | 6 (0.7%) | 9 (0.7%) | 15 (0.7%) |
| # of positive *Ae. aegypti* females with coinfection CHIKV–ZIKV (% female tested) | 3 (0.4%) | 0 (0.0%) | 3 (0.1%) |
| Fraction of each virus to all ABV positive mosquitoes (CHIKV, DENV, ZIKV) | (94.3%, 1.4%, 6.4%) | (0.0%, 65.3%, 34.6%) | (79.5%, 11.4%, 10.8%) |
| # of houses with positive *A. aegypti* females (+) for any virus (% of female tested/ infested houses with females) | 25 (37.9%) | 18 (17.5%) | 43 (25.4%) |
| # of houses (+) CHIKV (% of female tested/ infested houses with females) | 16 (24.2%) | 0 (11.7%) | 16 (9.5%) |
| # of houses (+) DENV (% of female tested/ infested houses with females) | 0 (0.0%) | 12 (5.8%) | 12 (7.1%) |
| # of houses (+) ZIKV (% of female tested/ infested houses with females) | 5 (7.6%) | 6 (37.9%) | 11 (6.5%) |
| # of houses (+) CHIV + ZIKV (% of female tested/ infested houses with females) | 1 (1.5%) | 0 (0.0%) | 1 (0.6%) |
| # of houses (+) CHIKV + DENV (% of female tested/ infested houses with females) | 2 (3.0%) | 0 (0.0%) | 2 (1.2%) |
| # of houses (+) with mosquito coinfection (CHIKV/ZIKV) (% of female tested/ infested houses with females) | 1 (1.5%) | 0 (0.0%) | 1 (0.6%) |

**Table 2. Number of anatomical structures of *Ae. aegypti* mosquitoes infected with either DENV, CHIKV and/or ZIKV, collected during the ABV transmission seasons of 2016 and 2017 in Yucatán, Mexico.** Percentages indicate the fraction of infection with each virus for each anatomical structure.

| Structure | DENV | CHIKV | ZIKV | CHIKV/ZIKV coinfection |
|---|---|---|---|---|
| Head | 1 (2.6%) | 33 (86.8%) | 1 (2.6%) | 3 (7.9%) |
| Body | 18 (13.7%) | 96 (73.3%) | 14 (10.7%) | 3 (2.3%) |

years, $X^2$ = 0.0024; $P$ = 0.96, Table 1). A similar trend among years was found for the rate of ABV infection in *Ae. aegypti* calculated by house (Table 1). Of the total ABV-infected females, 38 (22.9%) had evidence of infection in their heads; of those 33 (86.8%) were positive for CHIKV, 1 (2.6%) for ZIKV, and 1 (2.6%) for DENV (Table 2). Additionally, coinfections with CHIKV and ZIKV were detected in 3 (7.9%), which correspond to coinfections also detected in their bodies (Table 2).

Out of the total number of female mosquitoes, 81.3% were blood feed, at different blood feeding status (Sella's score), with 26.0% of them being fed withing 24-h of collection (Sella's score 2). The majority of positive females were blood engorged at the different blood feeding status (86.1%), with 34.3% freshly feed (Sella 2; Table 3). The remaining 33.1% of infected females were either unfed (19.3%—Sella 1) or gravid (13.2%—Sella 7) (Table 3). A 7.2% (n = 12) of the positive heads corresponded to positive bodies of female mosquitoes that were also classified with Sella score 2 (Table 3).

## Natural ABV infection rate of female *Ae. aegypti*

At the house level and when using the total catch of *Ae. aegypti*, ABV infections were detected in 43 houses (25.4%) out of 169 houses infested with female mosquitoes. In those 43 houses, ABV infections were divided as follows: 37.2% for CHIKV, 27.9% for DENV, and 25.6% for ZIKV (Table 1). Additionally, co-occurrence of mosquitoes infected with any of the three viruses was detected in 3 houses (7.0%) and 3 specimens of *Ae. aegypti* mosquitos co-infected with CHIKV and ZIKV were found in a single house (2.3%) (Table 1). The median of infected mosquitoes per positive houses was 1 (interquartile range [IQR] = 4–1). The distribution of positive females per house varied by virus, and for CHIKV was highly skewed with a maximum of 25 CHIKV infected *Ae. aegypti* in one house (Fig 1). The high overdispersion was further evidenced by the finding of 32.6% of houses contributing with 81.3% of the infected mosquitoes (Fig 1).

A significantly higher proportion of houses were found infected by any ABV in the high-density group (42.9%) compared to the low-density group (13.1%) ($X^2_{(df = 1)}$ = 17.6, P <0.001). When within household mosquito density was, a larger proportion of houses had mosquitoes

**Table 3. Distribution of virus infection among Sella scores, and their relationship with positive heads from *Ae. aegypti* collected during the ABV transmission seasons of 2016 and 2017 in Yucatán, Mexico.**

| Sella score | Sella score Interpretation | CHIKV | DENV | ZIKV | CHIKV/ZIKV | Total | Heads + |
|---|---|---|---|---|---|---|---|
| 0 | Unable to determine Sella | 1 (0.6%) | 0 (0.00%) | 0 (0.00%) | 0 (0.00%) | 1 (0.60%) | 1 (0.60%) |
| 1 | Empty abdomen | 27 (16.3%) | 2 (1.2%) | 3 (1.8%) | 0 (0.0%) | 32 (19.3%) | 5 (3.0%) |
| 2 | Engorged with intense blood | 47 (28.3%) | 6 (3.6%) | 4 (2.4%) | 0 (0.0%) | **57 (34.3%)** | 12 (7.2%) |
| 3 | Partially engorged with dark blood | 8 (4.8%) | 5 (3.0%) | 1 (0.6%) | 0 (0.0%) | 14 (8.4%) | 3 (1.8%) |
| 4 | Practically half full and half empty with dark blood | 10 (6.0%) | 0 (0.0%) | 2 (1.2%) | 2 (1.2%) | 14 (8.4%) | 4 (2.4%) |
| 5 | Less than half with black blood | 8 (4.8%) | 2 (1.2%) | 0 (0.0%) | 0 (0.0%) | 10 (6.0%) | 1 (0.6%) |
| 6 | Only anterior and ventral part with black blood | 14 (8.4%) | 2 (1.2%) | 0 (0.0%) | 0 (0.0%) | 16 (9.6%) | 5 (3.0%) |
| 7 | Abdomen full, with eggs or no visible blood | 14 (8.4%) | 2 (1.2%) | 5 (3.0%) | 1 (0.6%) | 22 (13.2%) | 7 (4.2%) |
| Total | | 129 (77.7%) | 19 (11.4%) | 15 (9.0%) | 3 (1.8%) | 166 (100%) | 38 (23.0%) |

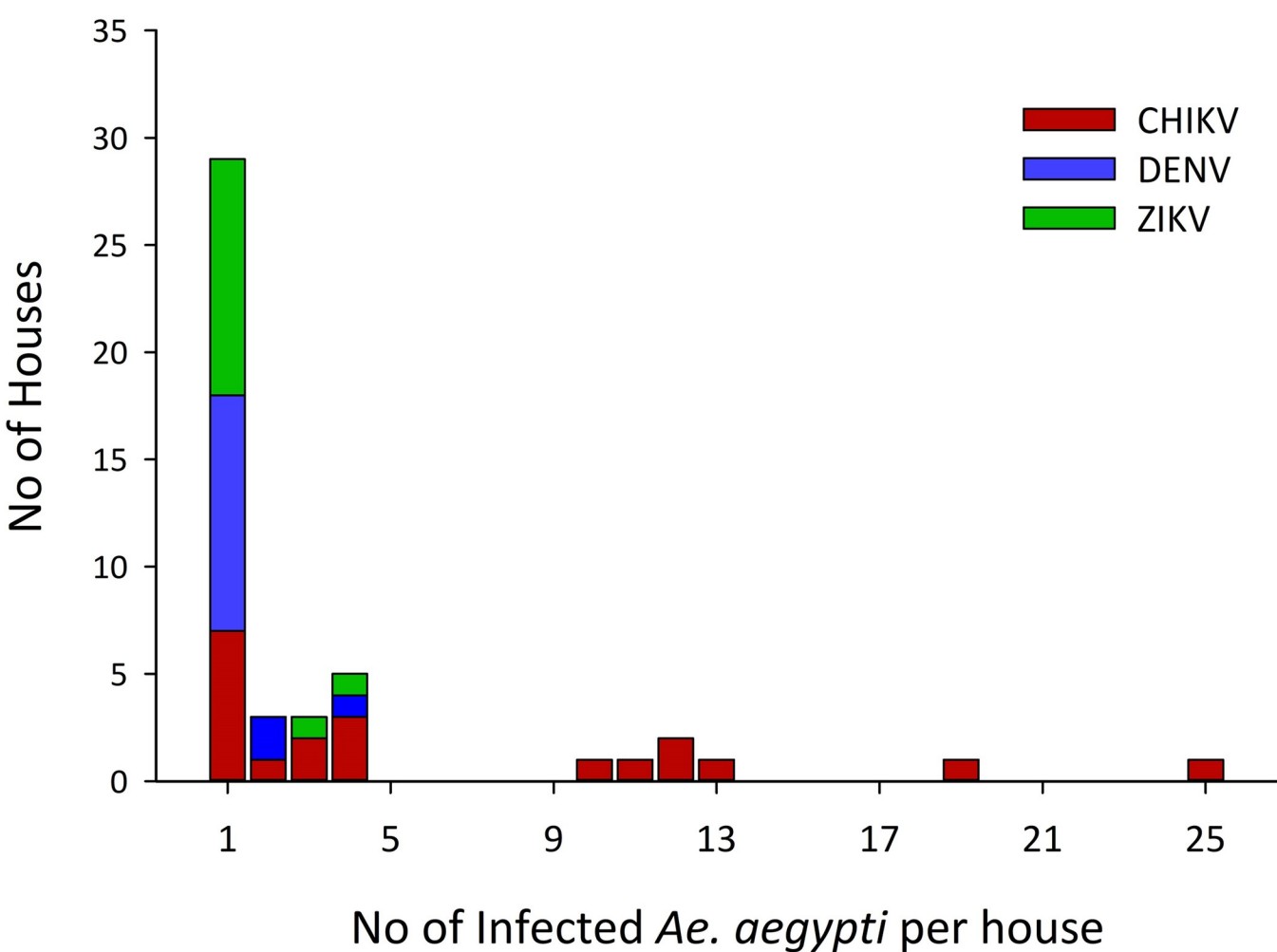

**Fig 1. Distribution of the number of female *Ae. aegypti* positive for CHIKV, DENV and ZIKV per house with positive mosquitoes collected in the ABV transmission seasons of 2016 and 2017 from Yucatán, Mexico.**

infected with CHIKV (18.6%) compared to DENV (12.9%) or ZIKV (8.6%); a 4.3% co-occurrence of infected mosquitoes with either virus was observed in high density houses. Comparatively, there was a similar proportion of houses with positive mosquitoes for each virus when mosquito density was low (Fig 2A). When analyzing mosquitoes with positive heads, only 3.0% were found in low-density houses while positive mosquito heads were found in 18.6% of high-density houses (Fig 2B). The probability of finding infected *Ae. aegypti* was significantly associated with total catch (binomial GLMM (Odds Ratio [95% CI]): 1.0 [1.0–1.1]), with houses having more than 40 *Ae. aegypti* females having a probability infection above 60% (Fig 3). When only considering infected female heads, no association with absolute density was found (1.0 [0.9–1.1]). Sella score did not have any significant association with infection for all adults or infected heads (S2 Table).

Fig 4 shows the sensitivity of Prokopack collections to the detection of ABV infected *Ae. aegypti* females. Performing a single 10-min Prokopack collection indoors led to a low (16.3%) sensitivity of detecting an ABV infected house (Fig 4A) or infected female (23.4%) (Fig 4B). The low sensitivity translated to each individual virus, both for houses (15.0% for CHIKV, 5.3% for DENV and, 25.0% for ZIKV) and individual mosquitoes (25.9% for CHIKV, DENV

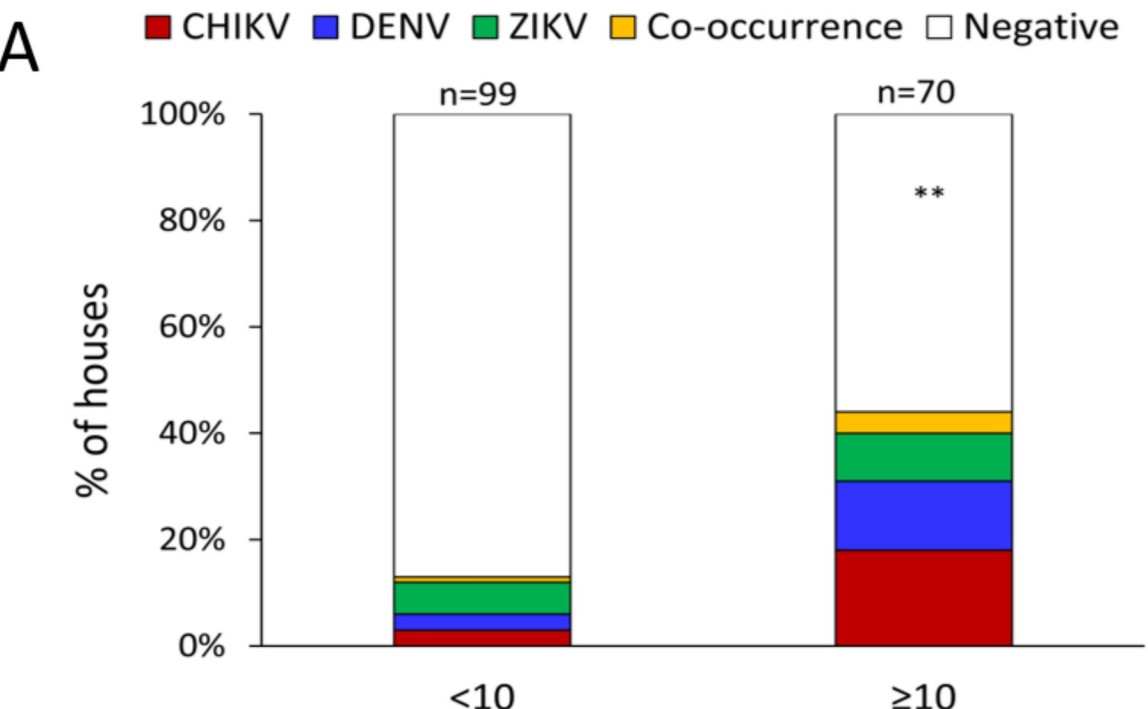

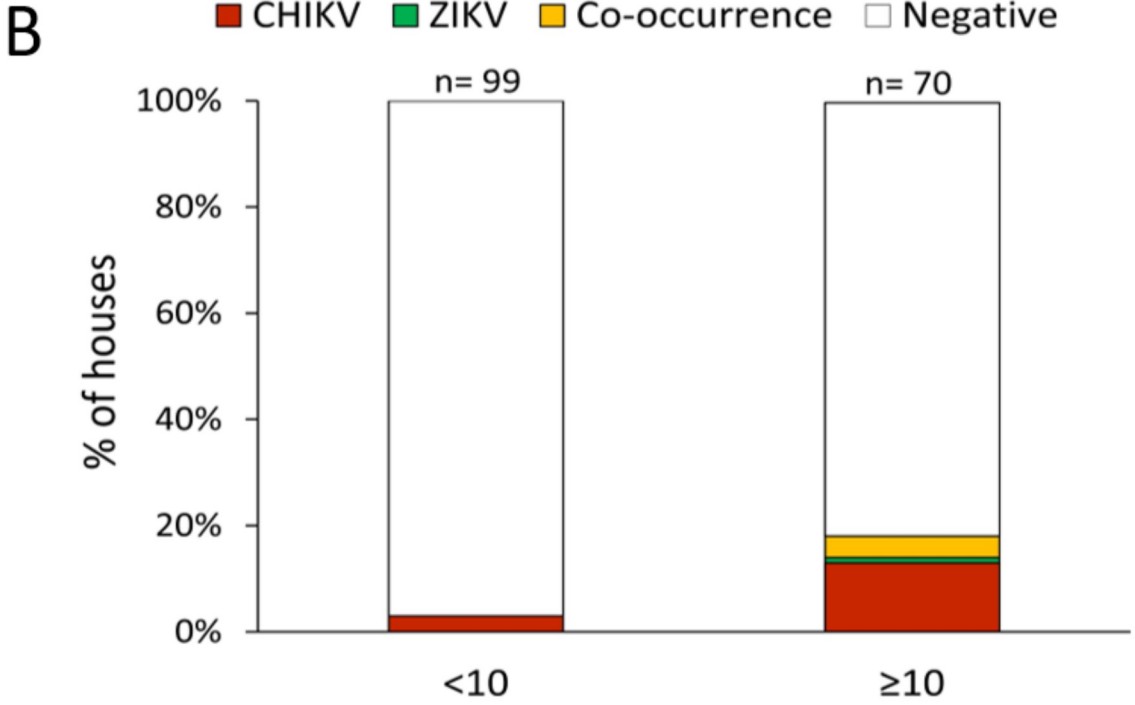

**Fig 2. Percentage of houses infested with female *Ae. aegypti* positive for any of the three targeted viruses in low-density (<10 total mosquitos per house, n = 98) and high-density (> 10 total mosquitoes per house, n = 70) premises, estimated from *Ae. aegypti* collected indoors during the ABV transmission seasons of 2016 and 207 in Yucatán, Mexico.** Panel A shows houses with positive bodies and heads and panel B shows the percentage of houses where only heads were positive. The variable co-occurrence contains percentages of houses where mosquitoes where positive for either virus within the same house, including three positive mosquitoes with coinfection between CHIKV and ZIKV.

5.3% and ZIKV 23.8%) (Fig 4). As collection time increased, the sensitivity of detection increased both for houses and mosquitoes, reaching an asymptote at ~120 min for any viral infection (Fig 4). Aggregating data from the first two sampling rounds (i.e. equivalent to performing a 20-min collection) led to an increase in household infection sensitivity (+16.3% for any adults, +15.0% for CHIKV, +26.3% for DENV and +15.0% for ZIKV; Fig 4A) and individual mosquito sensitivity (+17.5% for any adults, +16.8% for CHIKV, +26.3% for DENV and +14.3% for ZIKV; Fig 4B).

### Estimates of ABV transmission potential

The ratio of *Ae. aegypti* females to humans ($m$) increased significantly between the sampled mosquito density and the total catch (paired t-test = 6.4312, df = 199, p < 0.001; Fig 5A, S3 Table). At densities higher than 4 *Ae. aegypti* females, a GLMM predicted $m$ would surpass

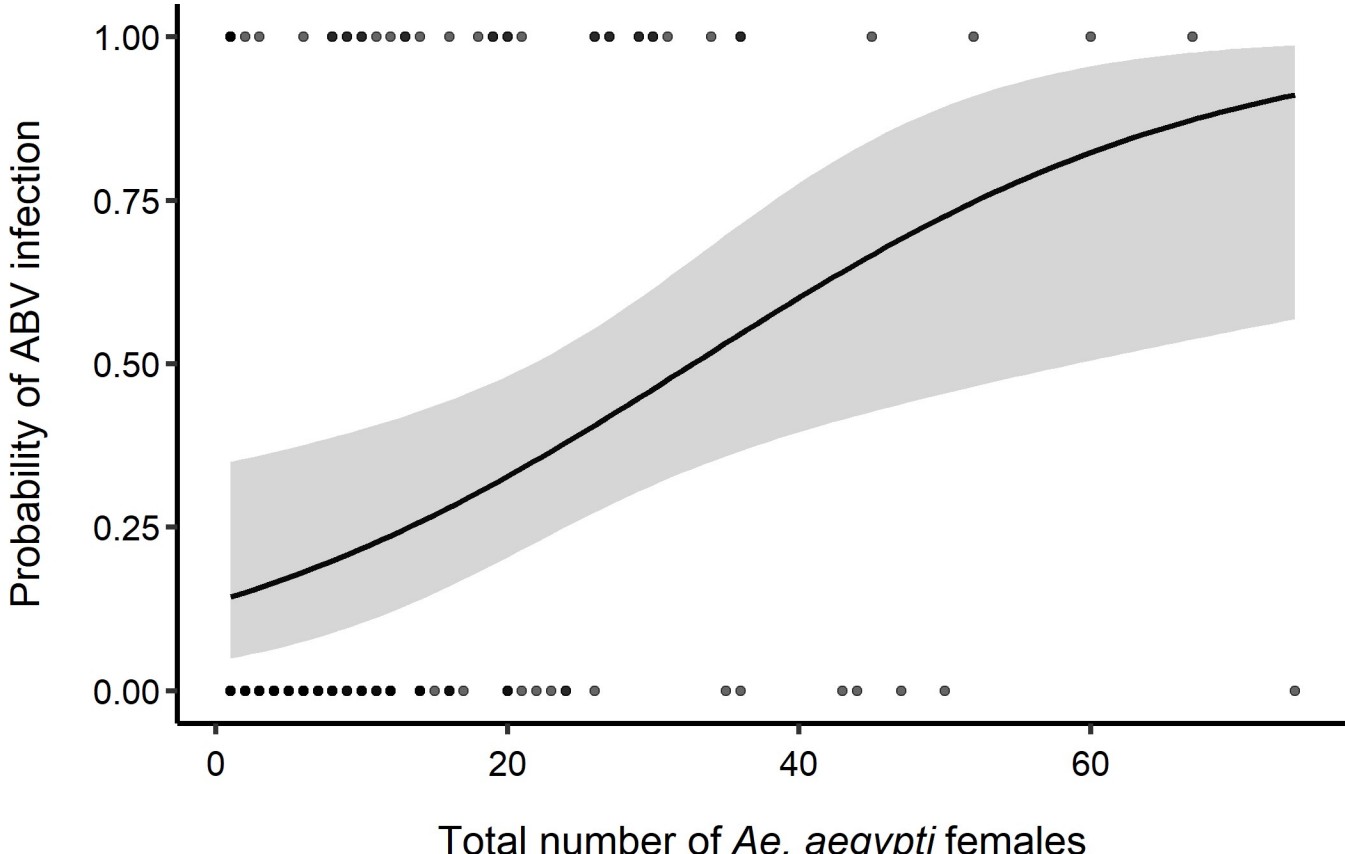

**Fig 3. Predicted probability of ABV infection in female *Ae. aeypti*.** Probability of detecting an infected female *Ae. aegypti* (0, uninfected, 1, infected) as a function of the total *Ae. aegypti* catch per house with evidence of recent arbovirus human infection, estimated from collections conducted during the ABV transmission seasons of 2016 and 2017 in Yucatán, Mexico. Solid line represents the mean prediction from a binomial generalized linear mixed effects model and gray band the 95% CI of the prediction, dots indicate the binomial data, with dark dots showing the occurrence of multiple (overlapping) observations.

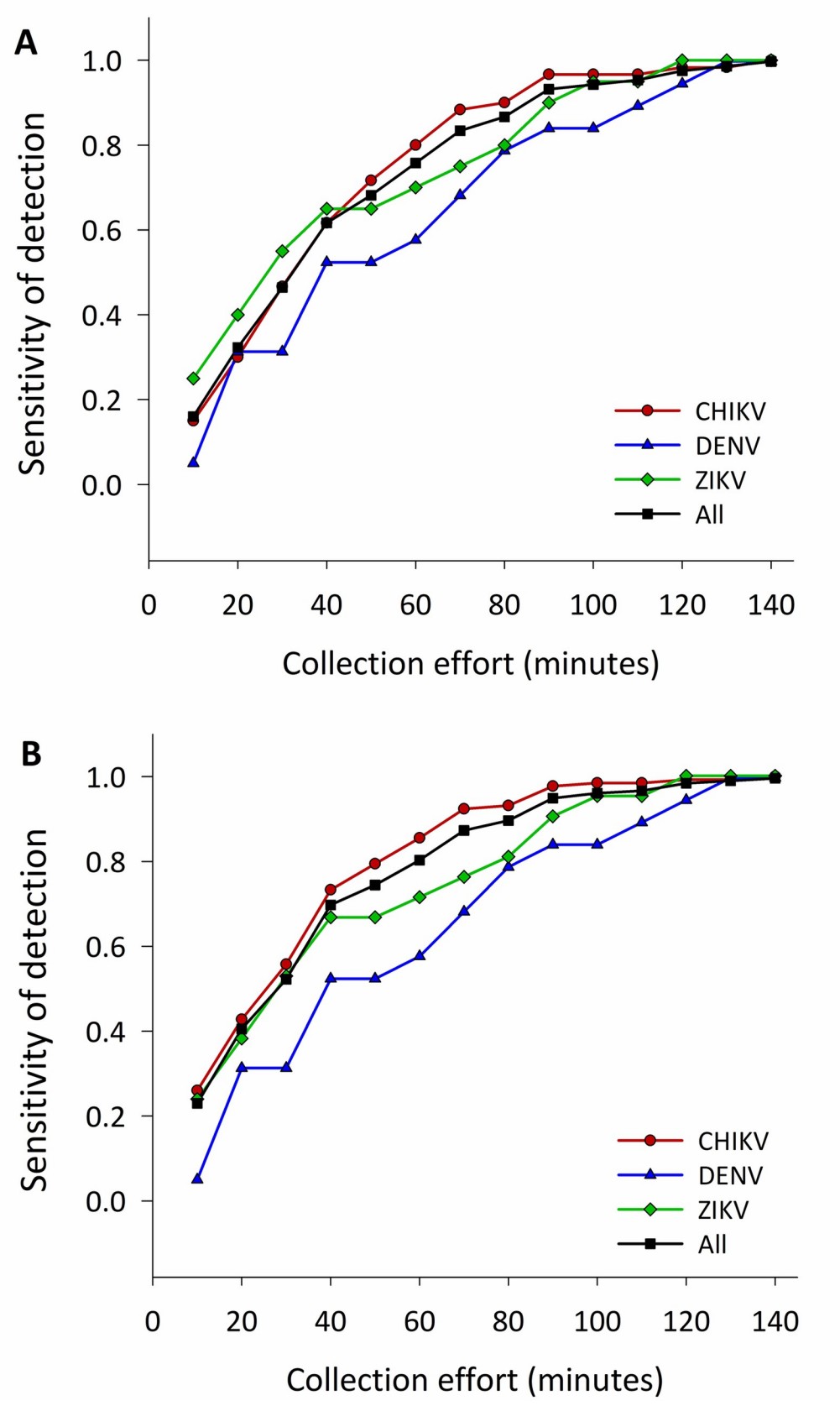

**Fig 4. Sensitivity of indoor adult aspiration to the detection of ABV-positive *Ae. aegypti*.** A) Cumulative probability of detecting houses with positive female *Ae. aegypti (*body and head) and B) cumulative probability of detecting positive female *Ae. aegypti* (body and head) for Chikungunya (CHIKV), Dengue (DENV) and/or Zika (ZIKV) in house as the collection effort increases in 10-min intervals. Estimates obtained from collections conducted indoors during the ABV transmission seasons of 2016 and 2017 in Yucatán, Mexico.

one; the maximum observed $m$ was 30 mosquitoes per person (Fig 5B). When averaged across all infested houses, mean EIR ranged between 0.04 and 0.06 infective bites per person per day, with estimates for total catch and sample being not statistically significant (paired t-test = -1.2988, df = 103, p = 0.1969) (Fig 5C, S3 Table). When only houses with infected *Ae. aegypti* were considered, mean EIR for the total catch increased to 0.28 infectious bites per person per day (Standard Deviation = 0.36; range = 0.01–1.5). Increasing total catch indoors lead to slight predicted variation in EIR (Fig 5D) likely due to the low infection rate (parameter $s$). ABV transmission potential, measured as mean VC, was significantly higher for the total catch than the sample (t = -2.6487, df = 103, p-value = 0.009) (Fig 5E). When scaled by total catch indoors, VC showed a significant increase from 1.0 below 20 *Ae. aegypti* females per house to 3.0 at density of 70 mosquitoes per house; the maximum VC estimate registered was 5.9 (Fig 5F, S3 Table). Large variability in feeding frequency (S2 Fig) influenced estimates of VC and EIR, for instance a house with a total catch of 60 had only 2 females *Ae. aegypti* with Sella's score 2, leading to a low estimate of parameter $a$.

## Nucleotide sequence analysis

Sanger sequencing confirmed with high fidelity the presence of three ABVs targeted by RT-PCR (S4 Table). High-quality reads matched perfectly or nearly perfectly (BLASTn search hit >90% identity) to CHIKV, DENV and ZIKV genomes published in NCBI GenBank. Consensus sequences were assembled for most samples sequenced for CHIK and ZIKV and will be used for future phylogenetic analysis. For DENV, ten single strand sequences confirmed DENV type 4 serotype as the circulating serotype (S4 Table). The virus identity of all positive heads matched the identity of the virus for the corresponding positive bodies (S4 Table).

## Discussion

CHIKV, DENV and ZIKV transmission risk appears to be correlated with the vector density and the number of infected mosquitoes at a coarse scale (entire cities, sub-national units) [25, 27], but such association between entomological indices and *ABV* incidence is generally inconsistent at the local level [27, 32, 51]. Our findings show that sampling bias in the quantification of vector density and in virus detection sensitivity as well as strong overdispersion in the distribution of infected mosquitoes may be important contributors to such inconsistency. We found that the sensitivity of routine Prokopack collections (10-min per house) in detecting houses with infected *Ae. aegypti* mosquitoes was below 25%. Furthermore, when infection was quantified in the total catch, approximately 80% of all infected mosquitoes were collected from ~30% of infested houses. Finally, infection in mosquitoes on a given year matched the dominant virus circulating in the human population. Taken together, such findings are relevant for the design of sampling schemes aimed at entomo-virological surveillance of *Ae. aegypti*, as it is evident that detecting infected mosquitoes will be a function of the sampling effort, the local abundance of *Ae. aegypti* and the intensity of arbovirus circulation. Furthermore, our findings may be transferable to other adult aspiration devices that operate under the same procedures and assumptions as the Prokopack (e.g., CDC back-pack aspirator and variants).

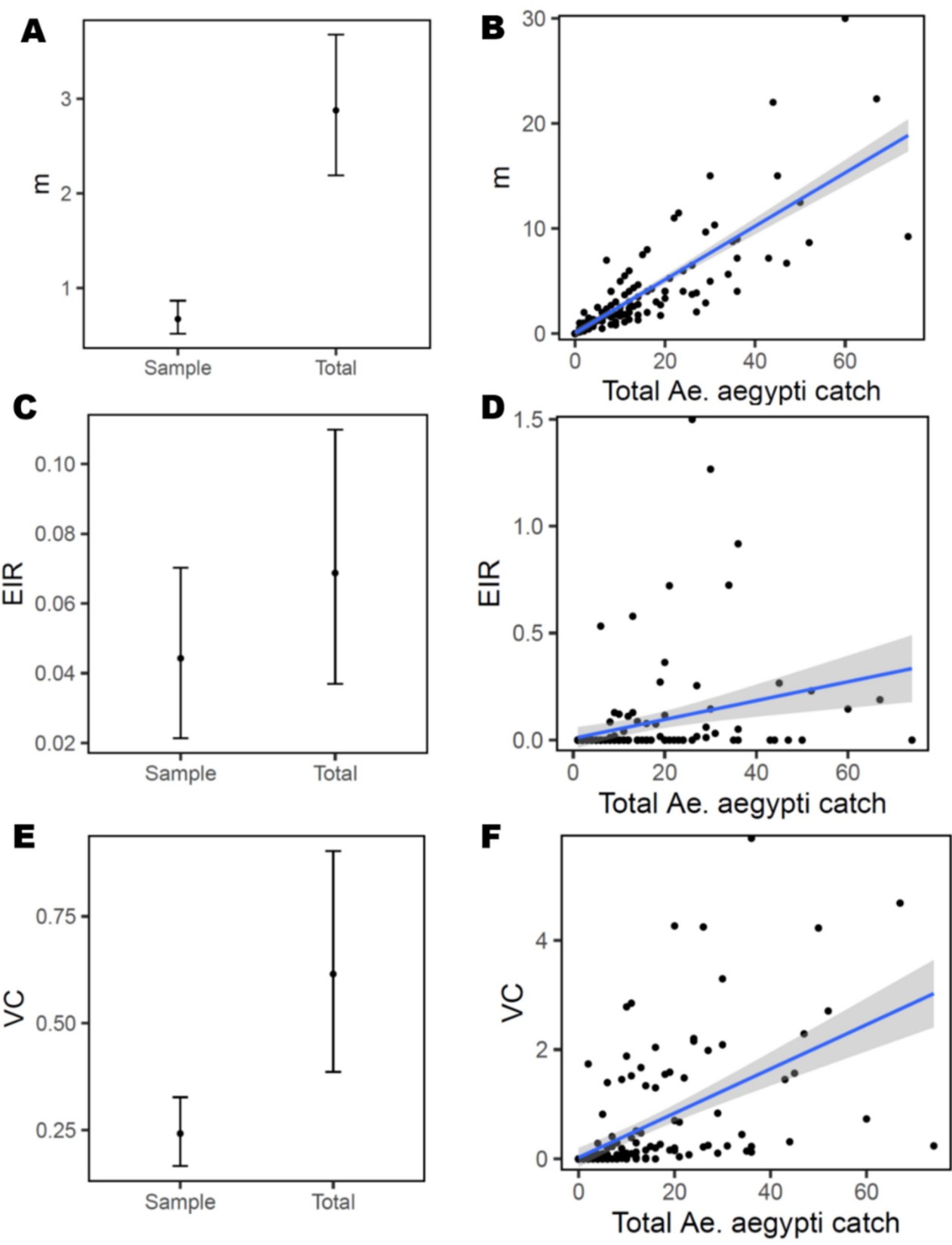

**Fig 5. Household-level estimates of ABV transmission potential.** The proportion of vectors per host (*m*), entomologic inoculation rate (EIR) and vectorial capacity (VC) were calculated per house and used to compare estimated between the first 10-min collection (sample) and the *Ae. aegypti* total catch (Total)(panels A, C, E). Panels B, D and F show the association between total *Ae. aegypti* female abundance per house, and estimates of *m*, EIR and VC, respectively. Lines show the fit and confidence interval of a generalized-linear mixed model fitted to the data (S3 Table).

In our previous study, we quantified that houses may harbor up to five times more adult *Ae. aegypti* than estimated during routine adult aspiration collections [18]. These data may help understand that the low apparent density of *Ae. aegypti* indoors, described in multiple studies (e.g., [15, 52]), may also be a function of the sensitivity of the collection method. The ability of *Ae. aegypti* to feed frequently (~1.5 days) and of distributing bites on some individuals more than others (aka., heterogeneous biting) [53–55] are considered the mechanisms compensating for the low *Ae. aegypti* density and human-mosquito rates [15]. Here we show that including the total *Ae. aegypti* population indoors (in our study, increasing the routinely sampled number by a factor of 5x) significantly elevates human—mosquito contacts and can have profound effects on estimates of natural infection and ABV transmission risk. Seven houses harbored more than 10 CHIVK infected females each, with one having up to 25 infected females. Multiple mechanisms could have been responsible of such overdispersion, including aggregation of bites on one or a few infected individuals, mosquito biting on multiple viremic visitors to the house, or the dispersal of infected mosquitoes from nearby premises. Limitations in our study design prevented us from identifying whether those houses with aggregated infection in mosquitoes had also one or multiple infected humans, limiting our ability to accurately determine the factors responsible for the large number of infected mosquitoes. As transmission of ABVs is shaped by the daily mobility patterns of humans [19, 56], any residents or visitors to such 'key locations' may experience a disproportionately high risk of infection. The asymmetric distribution of the number of infected mosquitoes could also be dependent on the availability of oviposition sites, some houses may be more prompt to harbor potential breeding sites which are related to the human behavior and housing characteristics. Evaluating the impact of observed total density of *Ae. aegypti* per house (which may well reach 100 females per house, [18]) on ABV transmission dynamics may help understand both the stability of virus transmission chains and the impact of vector control interventions focused on the indoor adult population. Several innovative strategies are being evaluated for their epidemiological impact on ABVs. Targeted Indoor Residual Spraying (TIRS) capitalizes on indoor resting behavior of *Ae. aegypti* (which primarily is found resting below 1.5 m and in dark surfaces) to deliver long-lasting residual insecticides that can significantly reduce vector density and dengue transmission [57, 58]. *Wolbachia* population replacement or suppression approaches rely on the release of genetically modified *Wolbachia* infected adults, which when mated to wildtype, uninfected adults, render the population incompetent to pathogen transmission or reduce adult female density, respectively [59]. Lethal ovitraps such as the *Aedes* gravid ovitrap (AGO) have shown important reductions in ABV prevalence and mosquito infection when deployed at high coverage [23]. Spatial repellents are volatilized pyrethroids that disrupt mosquito behavior and reduce human-mosquito contacts indoors, without apparent impact on population density [60]. All such approaches are dependent on an accurate characterization of the population density of the vector (for instance, release rates need accurate density estimates, repellency may not be effective at high vector numbers, residual effect may increase evolution of resistance at high densities, AGO traps may depend on estimates of vector density for their proper placement and coverage) and careful monitoring of their future implementation will require quantifying their effect on ABV infection. Quantification of sensitivity of existing methods (both to the detection of *Ae. aegypti* and to the detection of ABV-infected *Ae. aegypti*) is crucial to understand the entomological and epidemiological impact of

vector control. Furthermore, in the context of the current COVID-19 outbreak, methods such as indoor adult mosquito aspiration may not be easily implemented due to difficulties personnel my encounter in gaining access to the home environment. Aedes collection traps (e.g., GAT, BG sentinel trap, sticky ovitrap) would provide a viable alternative to indoor aspiration. Based on our findings we argue that future research should focus on calibrating such trapping methods to assess their sensitivity to detect *Ae. aegypti* and ABV-infected female mosquitoes.

A randomized controlled clinical trial will evaluate the epidemiological impact of TIRS in Mérida [41], with ABV infection in *Ae. aegypti* being quantified as a secondary endpoint. Our findings suggest that entomological collections with Prokopacks indoors should be conducted for more than the routine 10-minutes effort per house. Increasing the collection effort will increase the probability of detecting ABV infected *Ae. aegypti;* increasing effort to 20 minutes in houses where mosquitoes were found in the first 10-minute round would lead to a rapid increase in the sensitivity of Prokopack collections to the detection of ABV-infected mosquitoes. In the context of the TIRS trial, obtaining accurate measures of ABV infection in *Ae. aegypti* will lead to better estimates of the measured impact of the intervention, as it will allow quantifying what percent reduction in cases will be associated with a reduction in ABV infection in *Ae. aegypti* females. As other trials are implemented in the future, the consideration of the impact of an intervention on ABV infection in *Ae. aegypti* can be used to communicate vector control personnel the expected entomological effect of their actions.

Assessments of arbovirus infection in mosquitoes are commonly expressed as the prevalence of infections in pools of 15–30 individuals [27]. While MIR or MLR are commonly calculated, these indexes are prone to bias particularly if infection aggregates within a household [27, 32, 33]. Information from individually tested mosquitoes is more reliable, yet few studies have undertaking such expensive and time-consuming task. I In Mérida, RNA extracted from individual *Ae. aegypti* females collected in periods of high and low arbovirus transmission and tested by RT-PCR [30] led to estimates of DENV natural infection rate of <1% [30]. A similarly low DENV natural infection rate from individual mosquitoes was quantified both in Yogyakarta, Indonesia [61] and, in Ho Chi Minh City, Vietnam [62]. Particularly for DENV, our study found a similarly low natural infection rate (19/2,161 = 0.87%), in agreement with previous reports and consistent with the low number of reported dengue cases in the city (S1 Fig). Unlike such reports, which only focused on DENV, our study was conducted in the context of ongoing dengue transmission, and during CHIKV and ZIKV invasion of Mérida. Most of the positive *Ae. aegypti* had evidence of CHIKV infection during the first year of collection, which represented the second year post-CHIKV invasion in Mérida, leading this virus to contribute with three quarters of all infected mosquitoes to the overall 7.7% ABV natural infection rate. Our findings clearly show that, when multiple viruses co-circulate and transmit epidemically, infection rates may be influenced by the dominant virus. Surprisingly, in our study the high prevalence of infection by CHIKV in *Ae. aegypti* occurred in in 2016 during the emergence of ZIKA and a period when the most reported infections in Mérida were due to DENV (S1 Fig). Considering that most ABV infections go undetected to the public health system, either as asymptomatic or subclinical infections or for mild illness, may help explain the mismatch between high CHIKV infection in mosquitoes and the focus on ZIKV testing during this period of virus introduction into Mérida [14]. There are reports of early detection of CHIKV and ZIKV infection in *Ae. aegypti* from other states of Mexico prior to the detection of symptomatic cases [63, 64], which supports the known assumption that passive surveillance may fail to detect virus circulation in periods of low transmission.

We also found houses infested with mosquitoes positive for different viruses, suggesting the co- circulation of more than one virus within the area and even within the same house. CHIKV and ZIKV-positive mosquitoes were found in two houses, while CHIKV and DENV-

positive specimens were detected in one house. Additionally, coinfection of CHIKV and ZIKV was detected in three specimens within the same house. These data align with other studies that also reported the cohabitation of mosquitoes infected with different viruses within the same area or houses and the coinfection of two (or more) different viruses in individual mosquitoes. Cases of humans co-infected with multiple viruses have been reported in the Americas [65] and other regions [66–68]. Coinfections with all 3 arboviruses—CHIKV, DENV, and ZIKV—have also been reported [69, 70]. *Aedes aegypti* infected with more than one virus has also been detected, for example mosquitoes coinfected with ZIKV and DENV were detected in Manaus, Brazil, showing that ZIKV is preferentially transmitted over DENV when in coinfection [71]. Coinfection and transmission capacity of DENV/ CHIKV was also demonstrated through experimental infection of *Ae. aegypti* [72]. Notwithstanding, the epidemiological impact of multiple infections within the same vector or even multiple vectors within the same house is unknown.

In order to accurately confirm the detected ABV infection, we sequenced every PCR-positive sample. Our sequence reads positively confirmed the PCR results. In the case of DENV, we were able to typify the virial serotype as DENV-4. This result line up with previous results obtained from different work in the area where all four DENV serotypes were found circulating in Mérida, with DENV-4 being the predominant serotype in most years along with DENV-1 and DENV-3 serotypes [14, 73]. While PCR is a mainstream method for virus detection, ultimate confirmation of virus infection in mosquitoes should be done through virus isolation techniques using cell culture/suckling mice. Unfortunately, our field laboratory did not have the required BSL level and approvals to conduct virus isolation, as CHIKV requires a BSL3 facility. Despite this limitation, we consider our findings robust and tractable, because: a) sequencing of most PCR+ samples led to a direct match with the virus detected by PCR; and b) we found a 100% match between positive heads and positive bodies, indicating PCR conducted on the same individual led to similar results. We still consider our inability to isolate viruses as a limitation. Other limitations centered in our inability to obtain detailed information of the virus and number of individuals infected in each of the 200 houses. This knowledge gap, while not negative for the main conclusion of our study, limited our ability to identify factors influencing the distribution of virus infection in mosquitoes. Costs have limited our ability to conduct whole-genome sequencing of all the identified viruses, which would have helped fill some of the knowledge gaps about the epidemiology of ABVs in *Ae. aegypti*.

By individually testing mosquitoes and their body parts (head or abdomen) we unveiled important details about the process of infection and human-mosquito contacts in *Ae. aegypti*. The majority of recently blood fed females (Sella score 2) were positive for CHIKV (34.3%). We also detected 32 (19.3%) unfed (Sella score 1) infected females, which could be interpreted as females that had blood fed, digested the blood, fed again (within 24h of collection) and are ready for another gonotrophic cycle. Generally, mosquitoes are assumed to be infective when viral infection is detected in their head, which could indicate infection of the salivary glands. We found 38 female specimens with positive head, 86.8% of those were CHIKV-positive. Gravid females (Sella's score 7) with positive heads were also detected (7 specimens), but the majority of adults with infected heads had a Sella score of 2, indicative of mosquitoes that fed within the prior 24 h. As we are not sampling the same 'cohort' of adult mosquitoes, and visited the houses within one month of the report of a symptomatic case, we can hypothesize that those adult mosquitoes with positive heads acquired the viral infection from a viremic human, survived the extrinsic incubation period and just had a recent bloodmeal that likely led to virus inoculation on their human hosts. Quantifying the likelihood of such an important epidemiological event from our raw data would be very speculative, that's why we used our data to calculate indices of transmission potential or risk (VC or EIR). We found that transmission

potential (VC) was sensitive to the total density of mosquitoes collected, whereas transmission risk (EIR) was sensitive to the detection of infected mosquitoes. We acknowledge that metrics like VC may be sensitive to assumptions of heterogeneous biting, leading to a likely underestimation of VC, but our goal with its calculation was to empirically evaluate how VC is influenced when it is calculated using the sample versus the total catch of *Ae. aegypti*. Our analyses indicate that when *Ae. aegypti* total density is calculated, a significant association with the two measures of ABV transmission exists. Such findings highlight the relevance of accurate estimates of vector density and infection rates, and emphasize the value of studies quantifying the sensitivity of detection of *Ae. aegypti* and ABV infection in *Ae. aegypti* for informing the selection of any vector surveillance method.

## Supporting information

**S1 Fig. Number of clinical confirmed cases between 2015 and 2018 in Mérida, Yucatán. Mexico.** Data was obtained from SINAVE database, number of cases caused by CHIKV in 2015 was obtained from Méndez et al. 2017 [74]. Axis Y (Number of confirmed cases) is presented in Logarithmic scale. Dots on top of each bar represent the year of mosquito collection.
(TIF)

**S2 Fig.** Distribution of human biting rate (a) by house.
(TIF)

**S1 Table. Description and characteristics of real-time RT-PCR primer/probe sets used to target CHIK, ZIKV and DENV virus.**
(DOCX)

**S2 Table. Odd-ratio and 95% CI of the relationship between Sella scores and positive heads or positive bodies in *Ae. aegypti* collected from Yucatán, Mexico. No statistically significance was detected.**
(DOCX)

**S3 Table. Model fits for the association between entomologic inoculation rate (EIR) or vectorial capacity (VR) and total catch of female *Ae. aegypti* indoors.**
(DOCX)

**S4 Table. List of arbovirus-positive sequences that were used to confirm infection in collected mosquitoes from Yucatán, Mexico.**
(DOCX)

**S1 Data. Original dataset (in Excel format) with information per house (anonymized) of the number of *Ae. aegypti* collected, their infection status, the attributes of each house, and the data used to calculate EIR and VC.**
(XLSX)

## Acknowledgments

We thank the residents of Mérida, Yucatán, for kindly allowing us to conduct this important research.

## Author Contributions

**Conceptualization:** Oscar David Kirstein, Guadalupe Ayora-Talavera, Norma Pavia-Ruz, Pablo Manrique-Saide, Gonzalo M. Vazquez-Prokopec.

**Formal analysis:** Oscar David Kirstein, Guadalupe Ayora-Talavera, Daniel Chan Espinoza, Gonzalo M. Vazquez-Prokopec.

**Methodology:** Edgar Koyoc-Cardeña, Azael Che-Mendoza, Azael Cohuo-Rodriguez, Pilar Granja-Pérez, Henry Puerta-Guardo, Norma Pavia-Ruz, Mike W. Dunbar.

**Project administration:** Pablo Manrique-Saide, Gonzalo M. Vazquez-Prokopec.

**Writing – original draft:** Oscar David Kirstein, Pablo Manrique-Saide, Gonzalo M. Vazquez-Prokopec.

**Writing – review & editing:** Oscar David Kirstein, Guadalupe Ayora-Talavera, Edgar Koyoc-Cardeña, Daniel Chan Espinoza, Azael Che-Mendoza, Azael Cohuo-Rodriguez, Pilar Granja-Pérez, Henry Puerta-Guardo, Norma Pavia-Ruz, Mike W. Dunbar, Pablo Manrique-Saide, Gonzalo M. Vazquez-Prokopec.

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
