## [Decision Letter · Decision Letter 0]

16 Sep 2020

Dear Dr. Vazquez-Prokopec,

Thank you very much for submitting your manuscript "Natural Arbovirus Infection Rate and Detectability of Indoor Female Aedes aegypti from Merida, Yucatan, Mexico." for consideration at PLOS Neglected Tropical Diseases. As with all papers reviewed by the journal, your manuscript was reviewed by members of the editorial board and by several independent reviewers. In light of the reviews (below this email), we would like to invite the resubmission of a significantly-revised version that takes into account the reviewers' comments. 

We cannot make any decision about publication until we have seen the revised manuscript and your response to the reviewers' comments. Your revised manuscript is also likely to be sent to reviewers for further evaluation.

Sincerely,

Roberto Barrera, Ph.D.

Associate Editor

Nigel Beebe

Deputy Editor

Reviewer's Responses to Questions

**Key Review Criteria Required for Acceptance?**

**Methods**

-Are the objectives of the study clearly articulated with a clear testable hypothesis stated?

-Is the study design appropriate to address the stated objectives?

-Is the population clearly described and appropriate for the hypothesis being tested?

-Is the sample size sufficient to ensure adequate power to address the hypothesis being tested?

-Were correct statistical analysis used to support conclusions?

-Are there concerns about ethical or regulatory requirements being met?

Reviewer #1: yes, but what about PPE for sampling staff?

Reviewer #2: The methods are excellent with the exception of missing details outlined in the summary statements.

Reviewer #3: a. More information is needed in the study design. How were specific houses identified for sampling? Was it solely based on infection status of household members or did the authors try to randomize across households that had a history of symptomatic infection? Also, why only focus on households that had a history of symptomatic infection (this might become clearer when the study questions are added)? How were these homes distributed across space and time? Did you select homes to control for inter-home correlation by choosing homes that were spaced a certain distance apart? How many times were homes sampled (once vs. repeatedly)? I realize that these methods are likely associated with the reference provided in the text, but it would be nice to have some of this here to contextualize the discussion of the results.

b. I am unclear how the authors can justify that they are quantifying absolute vs. relative abundance of mosquitoes nor why this is an important distinction to make for the questions they are interested in addressing. To me relative abundance is absolute abundance (true abundance) multiplied by some estimate of capture efficiency. One can estimate absolute abundance by adjusting their measures of relative abundance by some imperfect detection rate. This has not been done in this study as currently written. In the current calculation of absolute abundance (sum aegypti over all of the 10 min intervals spanning 3 hr), the authors are assuming they have captured all of the aegypti in a given household. To me, this is just another measure of relative abundance (albeit likely a better one than just one 10 min interval) and if the authors want to get at absolute abundance than they need to adjust relative abundance by some metric of capture efficiency (which will go down with increasing effort). The authors may be able to get at this through the relationship between capture rate and time. If the authors do not want to do this, then they need to make their assumption that they have captured all of the aegypti in the house explicit in the manuscript and also discuss the implications for the interpretation of their results if this assumption is not upheld.

c. Statistical analyses: I imagine that the ability to detect arbovirus infection in mosquitoes will not only depend on mosquito densities within the house but also the number of household members that were symptomatic. Do you have this information to include in your model analysis? Also, all of the models constructed and the associated AIC scores should be included somewhere in the manuscript along with other metrics of model performance (residual analysis, etc.). Finally, I am not an expert on Bayesian approaches, but maybe these approaches would be appropriate for these data that explore, given a conditional probability that the household has arbovirus, the probability of detecting that arbovirus in mosquitoes and what factors (e.g. sampling effort and mosquito density) influence that probability?

d. Estimates of risk: Why use both EIR and VC? For the questions being addressed in this study as well as the implications for using data like these for targeted control, EIR seems a more relevant metric of risk. Further, the parameters comprising EIR can be directly estimated from the data generated in this study vs. VC, which has parameters that this study did not directly measure and likely change based on environmental conditions (mosquito survival rate and the extrinsic incubation period). Also, why is the probability of a mosquito becoming infected after biting an infectious person not included? This formulation of VC also has dimension issues. See: Massad Eduardo, Coutinho Francisco Antonio Bezerra. Vectorial capacity, basic reproduction number, force of infection and all that: formal notation to complete and adjust their classical concepts and equations. Mem. Inst. Oswaldo Cruz. 2012; 107 (4): 564-567. https://doi.org/10.1590/S0074-02762012000400022. I would recommend dropping VC unless further justification is provided and the potential limitations of VC are acknowledged in the discussion section.

**Results**

-Does the analysis presented match the analysis plan?

-Are the results clearly and completely presented?

-Are the figures (Tables, Images) of sufficient quality for clarity?

Reviewer #1: Yes

Reviewer #2: The results section is sufficient with the exception of suggested corrections in the summary statements.

Reviewer #3: Maybe? The questions of the study need to be clearly outlined along with the associated response variables. The tables and figures need work largely in clarifying the legends and some of the axes. These are noted in the uploaded document.

**Conclusions**

-Are the conclusions supported by the data presented?

-Are the limitations of analysis clearly described?

-Do the authors discuss how these data can be helpful to advance our understanding of the topic under study?

-Is public health relevance addressed?

Reviewer #1: Maybe consider alternative methods for measuring mosquito infections especially in COVID...Also, practicality of aspirating for large scale monitoring

SEE BELOW

Reviewer #2: Yes.

Reviewer #3: The conclusions are mostly supported by the data presented, but I did note when this was not the case throughout the discussion in the uploaded document. The limitations of the analysis are not currently described. The authors should include a paragraph in the discussion addressing study limitations, assumptions made, and implications for interpretation of their results. Public health relevance is addressed, but I have made suggestions on how to clarify this in the discussion.

**Editorial and Data Presentation Modifications?**

Reviewer #1: Minor

Reviewer #2: Minor comments:

Ln. 36. Replace ‘any’ to ‘at least one’

Ln. 40. At this point in the abstract it is not clear how ‘total catch’ and ‘number sampled’ are different. They sound like they are referring to the same thing but with different terms so try to better articulate the difference.

Ln. 42. You present two percentages and then say ‘respectively’, but it is not clear what these percentages refer to. 

Ln. 59. You use the phrase “entomo-virological approaches for virus surveillance”. Why not just swap this out with ‘arbovirus surveillance’? 

Ln. 68. This statement sounds awkward: “vector surveillance via entomo-virologic surveys”. Can you swap out with something like “virus testing of vector populations”. 

Ln. 89. This additional citation would be appropriate for this statement:

Ramirez et al. 2018. Searching for the proverbial needle in a haystack: advances in mosquito-borne arbovirus surveillance. Parasit Vectors. 11: 320.

Ln 325. I think ‘the’ should be replaced with ‘of’.

Ln. 357-359. You are claiming aggregation of bites based on these data but as discussed above, you are not presenting any data on the human infections. You could have had CHIKV pos mosquitoes in a home with a DEN patient. Or you could have had many CHIKV pos mosquitoes in a home with all occupants being CHIK patents.

Ln. 391. Check the papers below which might also provide DENV infection data for individual Ae. aegypti.

Rahayu et al. 2019. Prevalence and Distribution of Dengue Virus in Aedes aegypti in Yogyakarta City before Deployment of Wolbachia Infected Aedes aegypti. Int J Environ Res Public Health. 2019 May; 16(10): 1742.

Anders KL, Nga LH, Thuy NTV, Ngoc TV, Tam CT, Tai LTH, et al. (2015) Households as Foci for Dengue Transmission in Highly Urban Vietnam. PLoS Negl Trop Dis 9(2): e0003528. doi:10.1371/journal. pntd.0003528

In several locations scientific names (e.g. Ae. aegypti) need to be italicized. 

Fig. 3. Your legend could indicate what the dots at zero and 1 represent. I assume those are the raw binomial data with a sample at a given number per house that tested either pos or negative. But it looks like they have shades from light to dark and that is not explained.

Reviewer #3: Minor Concerns:

Line 39-40: Ae. aegypti needs to be italicized

Line 87-91: consider combining these two sentences with the following paragraph. Generally, for a paragraph to stand alone, there needs to be a minimum of three sentences. I also think these two sentences thematically align with the information in the following paragraph.

Line 114: “was detected 10 days...” in what? I am assuming field collected mosquitoes?

Line 148: (within 1 month) should be (within one month) – generally all numbers from 1-9 should be written out unless at the beginning of a sentence. I would go through the entire manuscript for this one, as there are multiple instances where this needs to be corrected.

Line 234: Based on the topic sentence of this paragraph (lines 231-232), VC and its definition should follow EIR and its definition before getting into the specifics of each metric.

Line 244: n = extrinsic incubation period should be the number of days on average it takes for the mosquito to become infectious (so, I am assuming this is 5 days). As currently written, this is defined as the extrinsic incubation rate (1/5 days), which would be incorrect.

Line 257-258: “Of the total ABV-infected females, 38 (22.9%) had evidence of infection in their heads; 33 (86.8%) of them were positive for CHIKV...” Do the authors mean 33 of the total ABV-infected females or 33 of the ABV-infected females with positive heads?

In general, the authors should make it really clear how the percentages presented throughout this manuscript are calculated. Especially in the associated tables (1-3), which should stand alone from the text, it is completely unclear what the percentages in parentheses are referring to.

Line 272: “follow” should read “follows”

Line 281: “When mosquito density was high…” I would clarify that this is “When within household mosquito density was high…”

Line 340: “considered a gold standard for indoor adult Ae. aegypti collections” – While this might be the case, I do not think this statement fits with what this sentence is trying to convey and I would recommend removing it. Especially since this study is not comparing sensitivity across different collection methods, this is confusing.

Line 345: “will be a function of the collection method used” – I would provide a reference here to support this statement as this study did not actually measure this. 

Line 348-350: “These data show that the low apparent…may also be a function of the sensitivity of the collection method.” I would clarify this sentence to reflect what this study actually measured – again, this study did not demonstrate the effectiveness of the Prokopack method relative to other collection methods, so starting the sentence with “these data show…” is somewhat misleading.

Line 355-357: “we evidenced its powerful epidemiological effect in the strong overdispersion of infection in collected mosquitoes.” I am not sure what the authors really are saying here. Please clarify. Also, can you really say that your data demonstrate heterogeneous biting? Or rather, that some households had more mosquitoes than others and the distribution of households with high infection rates also had high mosquito densities, so something to do with variation in housing infrastructure and mosquito habitat? I agree that people are not equally attractive, but I think other factors are likely driving the distribution of infected mosquitoes.

Line 358-359: see above comment. Also, do you know how many members of each household sampled were symptomatic for arbovirus infection?

Line 369: this sentence is not as clear as it could be. I suggest: “Wolbachia population replacement or suppression approaches rely on the release of Wolbachia infected adults, which when mated to wildtype, uninfected adults render the population incompetent to transmission or reduce adult female density, respectively.”

Line 372-376: This sentence needs clarification and this paragraph needs a better topic statement - I think I know what you are trying to say here...that the effective deployment and assessment of these technologies require good estimates of mosquito distributions across households, within household mosquito densities, and arbovirus infection rates. You should lead with this and then provide examples demonstrating why this is important.

Line 389: “bias” should be “to bias”

Line 393-396: The study design of the referenced study was different from the one in this study. It appears from the referenced study that they did not target efforts to households with confirmed symptomatic infection in household members. So, the prevalence would naturally be much lower than this study I would expect. So, I don’t understand the point the authors are trying to make here.

Line 406: “This data aligns” should be “These data align”

Line 413: “thought” should be “through”

Liine 418-419: “DENV-4 the predominated in the most years” is awkward. Consider “This result lines up with previous results obtained from different work in the area where all four DENV serotypes were found circulating in Merida, with DENV-4 being the predominant serotype in most years along with DENV-1 and DENV-3 serotypes.”

Line 424-426: Well...there is the salivary gland barrier that needs to be breached and can be a bottleneck in the infection process...so just because you see it in the head does not mean it is in the saliva...I would be careful here - maybe acknowledge this and then state that it is likely that these mosquitoes are infectious.

Line 435: I would argue that VC in this sense is not as good as EIR because multiple parameters you assume values for and these will be sensitive to environmental conditions.

Table 1: from the table legend, it is unclear how the percentages in the parentheses are calculated. 

Table 2 figure legend “either” ??? remove? Also, the are the percentages in this table reflecting the number of mosquitoes with positive heads out of total number of mosquitoes infected? Captured?

Table 3: again, how are the percentages calculated – tables and figures need to stand alone. I am assuming these are the number of mosquitoes with positive heads divided by the number of mosquitoes captured with each Sella score?

Figure 2 – maintain the same colors for each virus across each panel A and B – also line 697, 3 should be three

Figure 3 – the y-axis is confusing…it reads like the number, when this is really a probability of detection of infected females

Figure 4 – what is the difference between A and B?

Figure 5 – sample vs. total – do you mean first 10 min interval vs. the sum total across all 10 min intervals?

**Summary and General Comments**

Reviewer #1: This paper describes surveillance inside houses for Aedes born viruses in Merida Mexico. In particular it provides excellent quantitative data on the incidence of viremia mosquitoes in particular CHIKV, ZIKAV and DENV infections during concurrent outbreaks of these diseases. This is a very welcome addition to the literature. This data is then used to calculate indices of vectorial capacity for this important arbovirus vector. I think this is a very significant piece of research and should be published.

I do have a few comments that I would like the authors to consider and hopefully will improve the quality of the manuscript. While the use of aspirator collections particularly to collect mosquitoes indoors is a gold standard, there are some practical considerations that the authors must consider. In particular, during outbreaks of the coronavirus, it is all but impossible to enter people’s premises. While this hopefully it is but a temporary issue (I hope!), I still think it warrants I mentioned by the authors. Also some of the practical considerations of having to do with repeated collections within a house in order to detect aren’t there ways that this could be optimized? Other sampling methods could be used at least to provide information on virus incidence in the mosquito population, and to to measure the impact of a large scale intervention such as TIRS. This would include use of gravid Aedes traps to capture adult females which has been done in several other studies. Finally I think you should discuss the fact that Ae. aegypti is known to take multiple feeding and indeed biting attempts to obtain a bloodmeal. How does this affect VC/EIR?

Discuss implications of PCR antigen vs live virus detection.

I think this is a really nice piece of work and it’s great to see some actual field collections including multiple infections of viruses within Ae. aegypti. Well done.

Also please see other comments on the attached PDF.

Reviewer #2: Overall comments:

The authors published a study in 2019 estimating absolute indoor density of Aedes aegypti using sequential sampling of aspirators. This current study now builds on that prior work by reporting the virus testing results for these mosquitoes sampled from households with positive human cases as well as using these data to estimate the entomological inoculation rate and vectorial capacity. These are very unique data and the authors executed a very challenging study to gather these data. I’ll present a few discussion points below in hope of improving the manuscript.

These infection levels appear abnormally high. It would be helpful for the discussion to compare these results to other studies. Obviously this current student was targeting homes within a month of symptomatic humans infected with one of the three viruses. Virus infection studies in Ae. aegypti tend to report very low infection rates, especially for surveillance traps unrelated to locations of human cases, which is hard to compare to the current study. In your case you present an infection prevalence of 3.8% of your individual female Ae. aegypti being positive for CHIKV. The paper below is a good example of a study targeting Ae. aegypti collections by aspirators from inside the homes of human patients and they still only had 6 of 644 mosquitoes positive (0.9%). The current study appears to have very high infection levels in the collected mosquitoes so more explanation of this would help.

Anders KL, Nga LH, Thuy NTV, Ngoc TV, Tam CT, Tai LTH, et al. (2015) Households as Foci for Dengue Transmission in Highly Urban Vietnam. PLoS Negl Trop Dis 9(2): e0003528. doi:10.1371/journal. pntd.0003528

Figure S1 is very valuable as this understanding of virus circulation in the human populations at the community level is necessary to help interpret these mosquito infection results. The figure shows the total number of cases in 2015-2018. I think line 150 shows that the surveillance of cases and mosquito sampling occurred in 2016-2017. It would help if the methods could clarify when mosquitoes were sampled, including the months of collection. The tables and figures (e.g. Table 1, Fig. 1), could also include the year of collection in the table legend just for clarity. 

As I review the results, it seems as though all the data are grouped together ignoring the temporal component of when the samples were collected. This is unfortunate because the Fig. S1 shows very high variation in the amount of human cases for each virus in each year. A way to help resolve this would be if Table 1 could go from a single column of data as currently written to three columns (e.g. 2016, 2017, Total). The justification for paying more attention to when the samples were obtained is that given that 3/4ths of the positive pools were with CHIKV, we would assume most of those were from earlier in the sampling compared to later. In the discussion you say CHIKV was found in mosquitoes when most reported human infection was ZIKV. This further emphasizes the need to better present these data.

I also predict that comparing the high CHIKV in the mosquitoes there will still be a lack of correlation to what the SINAVE database reported for the community of Merida, Yucatan. Meaning the vast majority of human cases in 2016-2017 were not CHIKV so it is puzzling how this study documented so many CHIKV infections in mosquitoes. Your discussion trys to explain this but to help resolve, another improvement to this study would be to incorporate which virus was responsible for the human cases in the homes that were sampled. The methods explain that the Ministry of Health provided the addresses for human cases which was then sampled within 1 month of the patients being symptomatic. Is it possible to obtain the additional details regarding which homes had which viruses present when the mosquitoes were collected? You already know it was one of three viruses so knowing which virus specifically would likely avoid IRB issues with humans subjects (plus IRB already reviewed this study). While presenting all these mosquito infection data it would be much more valuable to know which virus was in the occupants of the household. For example, a virus that matches the mosquito and the household would suggest the occupants of the household resulted in exposure. Alternatively if they don’t match, it could suggest visitors to the home could have been infected. This would especially be interesting to know for the mosquitoes co-infected with 2 viruses.

Were vector control activities happening during the course of this study? That would help to know if these data generated here are during epidemics with no control, moderate control, intensive control, etc. The sampling focused on indoor populations of Ae. aegypti so I assume sampling of outdoor populations would have been different, especially in the presence of outdoor control efforts.

Table 3 is an interesting perspective looking at virus infection in bodies relative to heads for the positive individuals. It appears that 3% of the heads of unfed (parous and nulliparous) females with positive bodies were also infected. Then 7.2% of the heads with a fresh blood meal were positive. But what does not make sense is that you would expect that as an infected blood meal digests, the virus would disseminate in a portion and eventually reach the head. So as the sella score increases, ending at a gravid female, you would expect the heads to have a higher percent infection than unfeds. The percent infection of gravid females is 4.2% but the ones with a fresh blood meal less than a day old is 7.2%. This leads me to believe that there is a chance the specimens with a fresh blood meal could have had contamination from the virus in the blood meal resulting in the head testing positive. It would be good for the discussion to pay more attention to this.

Reviewer #3: This study focuses on quantifying Aedes aegypti density, and how sampling effort and mosquito density affect arbovirus detection rates, as well as estimates for arbovirus infection prevalence in sampled mosquitoes, the entomological inoculation rate, and overall vectorial capacity. This study provides a general approach and important data on the infection rates of indoor resting / questing mosquitoes that could be useful in targeting mosquito control and disease mitigation efforts. However, I have several major concerns that need to be addressed before this manuscript can be accepted for publication. 

Major Concerns:

1. Introduction: I think the last paragraph needs to include a succinct summary of the questions this study is interested in addressing. This is particularly important for understanding the specific study design and evaluating whether it is appropriate for the questions the authors are addressing. It is also important for understanding the response variables the authors have focused, the downstream comparisons made, and the choice of specific statistical analyses. From the information provided, I am assuming that the authors are trying to estimate how the probability of detection of arbovirus in mosquitoes within a home is affected by mosquito densities within the home and sampling effort given a high probability that arbovirus is present in that home. Right?

2. Methods: 

a. More information is needed in the study design. How were specific houses identified for sampling? Was it solely based on infection status of household members or did the authors try to randomize across households that had a history of symptomatic infection? Also, why only focus on households that had a history of symptomatic infection (this might become clearer when the study questions are added)? How were these homes distributed across space and time? Did you select homes to control for inter-home correlation by choosing homes that were spaced a certain distance apart? How many times were homes sampled (once vs. repeatedly)? I realize that these methods are likely associated with the reference provided in the text, but it would be nice to have some of this here to contextualize the discussion of the results.

b. I am unclear how the authors can justify that they are quantifying absolute vs. relative abundance of mosquitoes nor why this is an important distinction to make for the questions they are interested in addressing. To me relative abundance is absolute abundance (true abundance) multiplied by some estimate of capture efficiency. One can estimate absolute abundance by adjusting their measures of relative abundance by some imperfect detection rate. This has not been done in this study as currently written. In the current calculation of absolute abundance (sum aegypti over all of the 10 min intervals spanning 3 hr), the authors are assuming they have captured all of the aegypti in a given household. To me, this is just another measure of relative abundance (albeit likely a better one than just one 10 min interval) and if the authors want to get at absolute abundance than they need to adjust relative abundance by some metric of capture efficiency (which will go down with increasing effort). The authors may be able to get at this through the relationship between capture rate and time. If the authors do not want to do this, then they need to make their assumption that they have captured all of the aegypti in the house explicit in the manuscript and also discuss the implications for the interpretation of their results if this assumption is not upheld.

c. Statistical analyses: I imagine that the ability to detect arbovirus infection in mosquitoes will not only depend on mosquito densities within the house but also the number of household members that were symptomatic. Do you have this information to include in your model analysis? Also, all of the models constructed and the associated AIC scores should be included somewhere in the manuscript along with other metrics of model performance (residual analysis, etc.). Finally, I am not an expert on Bayesian approaches, but maybe these approaches would be appropriate for these data that explore, given a conditional probability that the household has arbovirus, the probability of detecting that arbovirus in mosquitoes and what factors (e.g. sampling effort and mosquito density) influence that probability?

d. Estimates of risk: Why use both EIR and VC? For the questions being addressed in this study as well as the implications for using data like these for targeted control, EIR seems a more relevant metric of risk. Further, the parameters comprising EIR can be directly estimated from the data generated in this study vs. VC, which has parameters that this study did not directly measure and likely change based on environmental conditions (mosquito survival rate and the extrinsic incubation period). Also, why is the probability of a mosquito becoming infected after biting an infectious person not included? This formulation of VC also has dimension issues. See: Massad Eduardo, Coutinho Francisco Antonio Bezerra. Vectorial capacity, basic reproduction number, force of infection and all that: formal notation to complete and adjust their classical concepts and equations. Mem. Inst. Oswaldo Cruz. 2012; 107 (4): 564-567. https://doi.org/10.1590/S0074-02762012000400022. I would recommend dropping VC unless further justification is provided and the potential limitations of VC are acknowledged in the discussion section.

3. Discussion: 

a. In the introduction and the discussion section, the authors make statements about the sensitivity of the Prokopack aspirators in detecting arbovirus infected mosquitoes and how this probability detection will be dependent on collection method. I would make sure that the language here is careful and reflects what the study actually examined. As currently reads, it seems as if the authors are arguing that the Prokopack is more sensitive to other collection methods (CDC backpack aspirator and other trapping methods). While this might be true, this is not what the study actually set out to measure. So clarify the language throughout to reflect how the factors measured (e.g. mosquito density, amount of effort, number of blood-fed mosquitoes, etc.) affect the sensitivity of detecting arbovirus infected mosquitoes. 

b. I think there are alternative (but not mutually exclusive) hypotheses that explain the overdispersion of infected mosquitoes across houses sampled. The aggregation of mosquito bites on a few attractive hosts could potentially explain this, but you are then assuming that mosquitoes can actively select certain households. As mentioned by the authors, aegypti does not disperse very far, so depending on how the sampled households in this study are located across space, there may be other explanations for the overdispersion. For example, variation in housing across the city could affect the accessibility of indoor habitat for questing and resting females, which would then affect the overall mosquito densities in a house. Further, what factors drive spatial variation in human arbovirus infections? Where the relatively few households that generated the most arbovirus infections in mosquitoes connected in any way or located spatially in a similar area? A broader discussion on potential mechanisms here is warranted.

c. I would add a concluding paragraph that summarizes the key implications the authors would like the readers to walk away with

PLOS authors have the option to publish the peer review history of their article (what does this mean?). If published, this will include your full peer review and any attached files.

Reviewer #1: No

Reviewer #2: No

Reviewer #3: No
---

## [Editor Report · Decision Letter 1]

10 Nov 2020

Dear Dr. Vazquez-Prokopec,

We are pleased to inform you that your manuscript 'Natural Arbovirus Infection Rate and Detectability of Indoor Female Aedes aegypti from Mérida, Yucatán, Mexico.' has been provisionally accepted for publication in PLOS Neglected Tropical Diseases.

Best regards,

Roberto Barrera, Ph.D.

Associate Editor

Nigel Beebe

Deputy Editor

---

## [Editor Report · Acceptance letter]

9 Dec 2020

Dear Dr. Vazquez-Prokopec,

We are delighted to inform you that your manuscript, "Natural Arbovirus Infection Rate and Detectability of Indoor Female Aedes aegypti from Mérida, Yucatán, Mexico.," has been formally accepted for publication in PLOS Neglected Tropical Diseases.

Best regards,

Shaden Kamhawi

co-Editor-in-Chief

Paul Brindley

co-Editor-in-Chief
